# EEG Network Analysis in Epilepsy with Generalized Tonic–Clonic Seizures Alone

**DOI:** 10.3390/brainsci12111574

**Published:** 2022-11-18

**Authors:** Dimitrios Pitetzis, Christos Frantzidis, Elizabeth Psoma, Georgia Deretzi, Anna Kalogera-Fountzila, Panagiotis D. Bamidis, Martha Spilioti

**Affiliations:** 1Department of Neurology, Papageorgiou General Hospital, 56403 Thessaloniki, Greece; 2Lab of Medical Physics and Digital Innovation, School of Medicine, Faculty of Health Sciences, Aristotle University of Thessaloniki, 54124 Thessaloniki, Greece; 3School of Computer Science, University of Lincoln, Lincoln LN6 7TS, UK; 4Department of Radiology, AHEPA General Hospital, School of Medicine, Faculty of Health Sciences, Aristotle University of Thessaloniki, 54636 Thessaloniki, Greece; 51st Department of Neurology, AHEPA General Hospital, School of Medicine, Faculty of Health Sciences, Aristotle University of Thessaloniki, 54636 Thessaloniki, Greece

**Keywords:** idiopathic generalized epilepsy, generalized tonic–clonic seizures alone (GTCSa), source localization, cortical connectivity, small-world propensity, integrated value of influence

## Abstract

Many contradictory theories regarding epileptogenesis in idiopathic generalized epilepsy have been proposed. This study aims to define the network that takes part in the formation of the spike-wave discharges in patients with generalized tonic–clonic seizures alone (GTCSa) and elucidate the network characteristics. Furthermore, we intend to define the most influential brain areas and clarify the connectivity pattern among them. The data were collected from 23 patients with GTCSa utilizing low-density electroencephalogram (EEG). The source localization of generalized spike-wave discharges (GSWDs) was conducted using the Standardized low-resolution brain electromagnetic tomography (sLORETA) methodology. Cortical connectivity was calculated utilizing the imaginary part of coherence. The network characteristics were investigated through small-world propensity and the integrated value of influence (IVI). Source localization analysis estimated that most sources of GSWDs were in the superior frontal gyrus and anterior cingulate. Graph theory analysis revealed that epileptic sources created a network that tended to be regularized during generalized spike-wave activity. The IVI analysis concluded that the most influential nodes were the left insular gyrus and the left inferior parietal gyrus at 3 and 4 Hz, respectively. In conclusion, some nodes acted mainly as generators of GSWDs and others as influential ones across the whole network.

## 1. Introduction

The electroencephalogram (EEG) is a non-invasive method of recording brain activity with high temporal and low spatial resolution. This technique has evolved from the measurement of post-synaptic potentials to the evaluation of neuronal networks. The EEG signal analysis includes network tools such as the estimation of source localization and measurement of connectivity [1]. The EEG network analysis permits the study of brain functions under various states such as resting [2], cognitive tasks [3], sleep [4] and emotions [5]. Additionally, it is a diagnostic method for neurological brain disorders, mainly altered consciousness [6], dementia [7], depression [8] and epilepsy [9].

Epilepsy is a brain disorder characterized by repetitive unprovoked seizures [10]. Graph theoretical analysis led to the perception of epilepsy as a network disorder rather than brain pathology restrained in specific sources [11]. The epileptic network is defined as functionally and structurally connected bilateral brain structures with interdependent activity. A manifestation of one such network is a subgroup of Genetic Generalized Epilepsies, the idiopathic generalized epilepsies (IGEs). IGEs are characterized by epileptic activity, which originates at a specific source and rapidly propagates to bilaterally distributed areas. The epileptic activity is attributed to the source and the network itself [12,13].

IGEs include the syndromes, childhood absence epilepsy (CAE), juvenile absence epilepsy (JAE), juvenile myoclonic epilepsy (JME), and generalized tonic–clonic seizures alone (GTCSa) [14]. IGEs constitute 15–20% of all epilepsies [15]. Interictal EEG reveals normal background activity and the epileptiform abnormalities of generalized spike-wave discharges. IGEs, by definition, have no structural etiology on conventional magnetic resonance imaging (MRI), except for the contribution of pathogenic gene variants in some cases [16,17]. Idiopathic generalized epileptic syndromes have distinct EEG findings in the frequency and the duration of the epileptiform discharges, which can help differentiate them [18].

Many theories regarding epileptogenesis in idiopathic generalized epilepsy have been proposed. Niedermeyer illustratively summarized the controversy regarding the role of the thalami [19]. There is evidence that the thalamus shows increased neuronal activity before generalized spike-wave discharges (GSWDs) [20] and is the primary driver from the initiation and throughout their propagation [21]. It has also been demonstrated that there is aberrant connectivity from cortical areas to the thalamus [22].

On the other hand, some researchers suggest that cortical areas such as the precuneus and the medial prefrontal cortex behave as crucial hubs in epileptogenesis. The thalamus seems to activate later [23,24,25]. Moreover, the epileptogenic network involves cortical areas beyond the frontal lobe and the frontal-thalamic pathway. The network pathology results from disrupted functional and structural connectivity [26], spreading across the resting-state networks, especially the default mode network [27].

The epileptogenesis in IGE could result from a pre-ictal state [19] of increased sensorimotor and decreased posterior connectivity [23]. This state potentially leads to cortical excitability, followed by the propagation of epileptic activity to the thalamus [28]. The thalamic reticular nucleus is responsible for the detected frequency of generalized spike-wave discharges [29]. EEG resting-state connectivity in IGEs has been well studied, whereas there is little data about EEG network connectivity in JME and CAE during GSWDs. Regarding the subsyndrome of GTCSa, the data are scarce.

This study aims to first define the network that takes part in the epileptogenesis of the spike-wave discharges in patients with the subsyndrome of GTCSa. Secondly, we purpose to elucidate the network characteristics and define the most influential brain areas. Finally, we intend to clarify their connectivity pattern in this sub-syndrome.

## 2. Materials and Methods

### 2.1. Patient Characteristics

Patients were recruited from the outpatient clinic of the First Department of Neurology, AHEPA Hospital. Inclusion criteria were (a) diagnosis of generalized tonic–clonic seizures alone, (b) age above 16, (c) normal brain MRI as diagnosed using conventional techniques. The patients were informed about the participation in this study and signed a consent form. The bioethics committee of Aristotle University of Thessaloniki approved this research (protocol number 1.74). The sample size was 23 patients with Generalized tonic–clonic seizures only. Patients’ characteristics were summarized in Table 1.

### 2.2. EEG Recording

The recordings were performed by an appropriately trained EEG technician using 19 channel EEG and supervised by a neurophysiology specialist. The electrode placement was carried out according to the International 10–20 EEG system. Four landmarks were used and specifically the nasion, inion, left and right pre-auricular points. The sample rate of the recordings was 256 Hz. The electrocardiographic activity and electrooculograms were recorded with bipolar electrodes. There were two reference electrodes attached to the mastoids and a ground electrode. The duration of each recording was 20 min and included hyperventilation as well as intermittent photic stimulation. The recordings took place in a quiet room with dim light and the patients were sitting in a comfortable armchair.

### 2.3. EEG Analysis

The Brainstorm freeware [30] was utilized for EEG preprocessing, epoch selection, head modeling, and source localization, as described in the following steps.

#### 2.3.1. Data Preprocessing

The EEG records cerebral activity and electrical activity known as artifacts. Several artifact sources, including the heart beats, ocular movements and muscle activity contaminate and suppress the information carried by EEG data. Therefore, filters can be used to remove artifacts from the EEG signal [31]. Low-pass filters reduce high-frequency variance that can be safely attributed to unimportant noise fluctuations. High-pass filters eliminate slow elements such as direct current changes upon which superimpose the faster signals of brain activity. Power line/industrial noise (50 or 60 Hz and its harmonics) that connect electrically or magnetically with the recording circuits frequently interferes with electrophysiological signals. As described in Chriskos et al. paper, additional notch filters centered at the harmonic frequencies could be implemented to provide more efficient power line cancelation. Although the low-pass filter with cut-off frequency at 50 Hz attenuates the power line harmonics, it cannot fully minimize their effects and an additional notch filter was implemented [32,33]. To remove spectral content irrelevant to brain activity, such as power line noise, the following third-order Butterworth filters were applied:High-pass filter/cut-off frequency at 0.5 Hz;Low-pass filter/cut-off frequency at 50 Hz;Band-stop filter/ranging from 47 to 53 Hz;Band-stop filter/ranging from 97 to 103 Hz;Common average re-referencing was used [34].

Epochs with biological artifacts such as body movements, ocular movements and eye blinks were excluded through visual inspection of the EEG, which was carried out by a neurophysiology specialist. The recordings were not preprocessed with Independent Component Analysis (ICA) [35] to avoid the rejection of clinically relevant data such as the generalized spike-wave complexes.

#### 2.3.2. Events

Global field power (GFP) was calculated with a Matlab based, custom script, to determine when electrodes recorded maximum energy during each spike. Events were defined as the time points corresponding to the midpoint of the ascending part of each spike [36]. GFP quantifies the power of a potential scalp field, considering all the electrodes similarly. Furthermore, GFP is reference-independent and defined as
(1)$$\mathrm{GFP}=\sqrt{\sum_{i=1}^{N}\frac{{\left(u_{i}-\bar{u}\right)}^{2}}{N}}$$
where is the voltage of the map at the electrode , is the mean voltage of all electrodes and *N* is the number of electrodes [37].

#### 2.3.3. Epoch Selection

For each subject, two sets of epochs were chosen by a neurophysiology specialist.

1. Epochs with generalized spike-wave complex (GSW) activity were known as a GSW epoch. The first spike of each GSW trace was used as the main event. Epochs included two seconds before and two seconds after the main event (4 s).

2. Epochs with resting-state (RS) activity were known as an RS epoch (4 s).

For each subject, the number of RS epochs was set equal to the number of GSW epochs.

### 2.4. Source Localization

#### 2.4.1. Forward Problem

Source localization firstly requires the solution of a forward problem. This refers to the estimation of a model which explains how the brain activity is measured from the sensor electrodes on the scalp. The solution to the forward problem requires an electrical source configuration, which represents the activated brain neurons, the coordinates of the sensor electrodes and the electrode alignment on the head model. The forward modeling process considers the various head tissues (white and grey matter, cerebrospinal fluid, skull bone, and skin) which express different conductivity of electrical activity [38,39].

In our study, we used the default 10–20 electrode positions supplied by the Brainstorm toolbox, which were converted to the Montreal Neurological Institute (MNI) space [40,41]. The head modeling was conducted through the Open MEEG Boundary Elements Method (BEM head model) based on the generic anatomy of International Consortium for Brain Mapping (ICBM) 152 atlas [42]. BEM solves the forward problem by estimating the potential values at the interfaces (scalp, skull, brain) and boundary (outer surface) of the model induced by a current source. The solution space is consisted of a 3-dimensional grid of 15,000 fixed dipoles oriented normally to the cortical surface [43].

#### 2.4.2. Inverse Problem

The inverse problem estimates the sources which account for the recorded potential at the sensor level. Several variables, including head-modeling errors, source-modeling problems, and EEG noise (external or biological), might affect how accurately a source can be found [44]. There is no unique solution to the inverse problem, hence, mathematical constraints prior to the source estimation are required [39].

The computation of the inverse problem was estimated using the Standardized low-resolution brain electromagnetic tomography (sLORETA) methodology. The localization method is based on normalized current density using the source and noise variances.

Noise covariance statistics were calculated from the recordings. One artifact-free epoch with resting-state activity was selected for each subject. These epochs were selected and preprocessed as described in Section 2.3.1. The duration of each epoch was ten seconds.

#### 2.4.3. Regions of Interest

Regions of interest (ROI) were defined as the area (voxel) with the maximum recorded energy during each event. Using the MNI coordinates, the ROIs were correlated to the surface of the anatomical regions using the Automated Anatomical Labelling atlas (AAL atlas) [45] with the xjview toolbox “http://www.alivelearn.net/xjview (accessed on 1 May 2022)”.

### 2.5. Cortical Functional Connectivity Analysis

Connectivity in the context of EEG refers to the analysis of the links between two or more EEG signals. EEG connectivity is calculated using activity potentials collected from the scalp. Functional connectivity regards the statistical correlations among activities of different brain regions. It can be evaluated either at the sensor level or the cortical level [46].

The raw sensor data were transformed into two sets of cortical data (GSW and RS), which contained the estimated values of power F of the sources during each epoch. Cortical connectivity was calculated using the HERMES toolbox [47], utilizing the imaginary part of coherence for the frequencies 1–12 Hz. The coherence is a complex number with an absolute value less than or equal to 1 and an angle which is correlated to the phase lag between the signals. The imaginary part of coherence reflects the interaction of sources and is insensitive to volume conduction [48]. For every patient, two matrices were created for each frequency. Each pair contained the average cortical connectivity among nodes during GSW and RS epochs at a specific frequency. These data comprised the weights of the undirected network.

### 2.6. Network Construction

In graph theory, a network is described as a collection of nodes or vertices and the edges or lines connecting them. The network can be represented in the form of a connection (or adjacency) matrix. Depending on the measurement method, brain nodes can be isolated neurons or whole brain regions. Edges may have weighted or binary values and may be directed or undirected. Numerous metrics can be used to define the topology of graphs. While global metrics represent the properties of the entire network, local metrics deal with the relationship of certain nodes to the network structure [49,50].

#### 2.6.1. Nodes

Nodes consisted of the surface of the anatomic areas, defined by the AAL atlas at “Section 2.4.3”, including the thalami.

#### 2.6.2. Adjacency Matrix

Network-Based Statistics Connectome (NBS) toolbox was used to estimate the connections which present with statistically significant probability comparing two states—GSW activity and RS activity. The following options were selected for the statistical model: design matrix—one group of subjects and two conditions; statistical test—*t*-test and the threshold value was set to 1 (T = 1). The number of permutations was set to 5000.

The output was a binary adjacency matrix network, where “1” corresponds to a statistically significant probability of rejection of the null hypothesis. Therefore, there is a statistically significant probability that two nodes connect. The opposite applies to value “0” at the adjacency matrix [51].

#### 2.6.3. Weighted Modified Connectivity Matrix

The binary adjacency matrix was transformed to the weighted connectivity matrix by substituting the statistically significant marker “1” with the edges’ actual connectivity value (calculated at step E). The connectivity value of the non-significant edges was set at zero. The result was one network per frequency with two sets of weights, one for GSW activity and one for RS activity.

### 2.7. Graph Theory Analysis

Graph theory [52] was used to determine specific network characteristics. The characteristic path length quantifies the ease with which information flows between brain areas. The weighted characteristic path length is defined as
(2)$$L^{w}=\frac{1}{n}\sum_{i\in N}\frac{\sum_{j\in N,\;j\neq i}d_{ij}^{w}}{n-1}$$
where *N* is the set of all nodes; *n* is the number of nodes; links (*i*, *j*) are associated with connection weights *w_ij_*; $d_{ij}^{w}$ is the shortest weighted path length between *i* and *j*. The mean clustering coefficient (mean C) measures the tendency of a network to create local circuits around individual nodes. It is related to the network’s ability to process information. The weighed clustering coefficient [50,53] is defined as
(3)$$C^{w}=\frac{1}{n}\sum_{i\in N}\frac{2t_{i}^{w}}{k_{i}\left(k_{j}-1\right)}\;$$
where *N* is the set of all nodes; n is the number of nodes; links (*i*, *j*) are associated with connection weights *wij*; *k_i_* is the degree of a node *i*; $t_{i}^{w}$ is the weighted geometric mean of triangles around *i*. Characteristic path length (L) and mean clustering coefficient were calculated for the frequencies 1–12 Hz. Further steps were conducted at the frequencies of 3 and 4 Hz based on a cohort study which demonstrated that the generalized spike-wave frequency ranged from 3.1 to 4 Hz [18]. 

The small-worldness in weighted networks was determined through small-world propensity (SWP) [53]. Below are the formulas for the calculation of SWP:(4)$$\varphi =1-\sqrt{\frac{\Delta_{C}^{2}+\Delta_{L}^{2}}{2}}$$
(5)$$\Delta_{c}=\frac{C_{latt}-C_{obs}}{C_{latt}-C_{rand}}\;$$
(6)$$\Delta_{L}=\frac{L_{obs}-L_{rand}}{L_{latt}-L_{rand}}\;$$
where *C_obs_* is network’s clustering coefficient, *L_obs_* is network’s characteristic path length, *C_latt_* and *C_rand_* are clustering coefficient of lattice and random network, respectively, *L_latt_* and *L_rand_* are the characteristic path lengths of lattice and random network, respectively. Values of *φ* greater than 0.6 indicate strong small-world propensity. The values of Δ*_C_* and Δ*_L_* correspond to the divergence of the metric from the regular and random network, respectively. Furthermore, contribution to deviation is defined as
(7)$$\delta =\frac{4a}{\pi }-1\;$$
where *α* = tan^−1^ (Δ*_L_*/Δ*_C_*) − 1 is the angle of the observed vector. This metric indicates whether Δ*_L_* or Δ*_C_* influence the SWP value [54].

The integrated value of influence (IVI) is a metric that combines six different connectivity measures (degree centrality, Cluster-Rank neighborhood connectivity, local H index, betweenness centrality and collective influence) to estimate influential nodes. The topological position of nodes does not influence the collaborating effect of these measures. The determination of IVI requires the calculation of two scores: the spreading and the hubness one. The spreading score is correlated to the ability of a node to spread information. The hubness score quantifies the effect of each node on its domain. The IVI is defined as
(8)$${\mathrm{IVI}}_{i}=({\mathrm{Hubness}}_{{\mathrm{score}}_{i}})({\mathrm{Spreading}}_{{\mathrm{score}}_{i}}).$$
(9)$${\mathrm{Spreading}}_{{\mathrm{score}}_{i}}=\left({\mathrm{NC}}_{i}^{'}+{\mathrm{CR}}_{i}^{'}\right)\left({\mathrm{BC}}_{i}^{'}+{\mathrm{CI}}_{i}^{'}\right)$$
(10)$${\mathrm{Hubness}}_{{\mathrm{score}}_{i}}={\mathrm{DC}}_{i}^{'}+{\mathrm{LH}}_{{\mathrm{index}}_{i}}$$
where ${\mathrm{NC}}_{i}^{'}$, ${\mathrm{CR}}_{i}^{'}$, ${\mathrm{BC}}_{i}^{'}$, ${\mathrm{CI}}_{i}^{'}$, ${\mathrm{DC}}_{i}^{'}$, ${\mathrm{LH}}_{{\mathrm{index}}_{i}}$ are normalized neighborhood connectivity, ClusterRank, betweenness centrality, collective influence, degree centrality and local H index of node i, respectively.

The IVI was calculated for 3 and 4 Hz based on the weighted undirected adjacency matrices created at step E3 [55]. This step was conducted using RStudio software [56] and the R package “influential” [55,57]. Nodes with an IVI score higher than the mean value plus one standard deviation were arbitrarily defined as those with more substantial influence.

### 2.8. Statistical Analysis

Statistical analysis was conducted using the IBM SPSS Statistics software version 20. Firstly, the normality assumption of the differences between values of characteristic path length for both states and all frequencies was checked. Because of the size sample, the Shapiro–Wilk test was used. If the variable had a normal distribution, a paired *t*-test for each frequency was conducted to find statistically significant differences. On the contrary, the Wilcoxon signed-rank test was applied if the variable did not follow normal distribution. Two factors repeated measures ANOVA test [28] was used to examine how the variance differed between the GSW and RS states for all frequencies collectively. A two-factor repeated measures ANOVA test was used even though the assumptions were not fulfilled for two frequencies. The method is robust even in such cases [58,59]. Sphericity was checked with Mauchly’s test, and Greenhouse–Geisser correction was applied to examine whether there was any interaction between state and frequency or not. Finally, Tukey’s test and Bonferroni correction were used to find statistically significant differences.

### 2.9. Connectivity Pattern among Influential Nodes with Low Density at 3 and 4 Hz

We repeated the NBS analysis with a much stricter value of threshold (T = 3.5) to visualize and examine the significance of connections among influential nodes during the GSW activity. The statistical test and the permutations remained unchanged [51]. The connectivity pattern was visualized using BrainNet Viewer Toolbox [60]. The Figure 1 summarizes the methodology that was applied in this research.

## 3. Results

A.Source localization of spikes

Collectively, among 23 subjects, 263 spike-wave complexes were analyzed as discussed in section Methods. Table 2 contains the results of source localization, which are illustrated in Figure 1. Most sources belonged anatomically to the frontal and limbic lobes. Specifically, the superior frontal gyrus, middle frontal gyrus and anterior cingulate were the most common sources of spikes.

B.Graph theory analysis

B1.Characteristic path Length

The difference in the variable between the states followed a normal distribution regardless of the frequency. For all frequencies 1–12 Hz, there was a statistically significant reduction in the mean value of characteristic path length in the GSW state compared to the resting state (Figure 2). There was a statistically significant difference between the two states regardless of the frequency with F (1, 22) = 168,463 and *p* < 0.001. More specifically, there was a decrease in the mean value of the variable L in the GSW network of the order of 4.208 (95% confidence interval: 3.536 to 4.880), regardless of the frequency. With the Bonferroni correction, the *p*-value remains less than 0.001. For the variable L, no statistically significant interaction was observed between the 12 frequencies and the two states of the subjects regarding their respective measurements, F (4.824, 106.118) = 1.060 and *p* = 0.386, and partial η^2^ = 0.046. The observed decrease in the characteristic path length is translated to a more effortless flow of information in the GSW state.

B2.Mean clustering coefficient (mean C)

The parameters of the differences in the variable mean C followed a normal distribution, except for the frequencies of 5 Hz and 12 Hz. The *t*-test for paired observations (paired *t*-test) was used to detect a possible statistically significant difference between the two states in each of the ten frequencies. Respectively, for those frequencies that did not follow a normal distribution, the Wilcoxon signed-rank non-parametric test was applied. Examining the results of the paired *t*-test, we observed that at the frequencies 1, 2, 3, 4, 6 and 10 Hz, there was a statistically significant increase in the mean value of the variable mean C in the GSW state, after the application of Bonferroni correction. At 5 Hz, there was a statistically significant increase in the median comparing the two states. Sphericity was tested through Mauchly’s test. Because the value of *p* was < 0.05, this criterion was not met, and the Greenhouse–Geisser correction was used. There was a statistically significant interaction between the two factors (state and frequency) F (5.002, 110.049) = 4.402 and *p* = 0.001. This is evident in Figure 3, as the two lines seem to differ a little more significantly in the frequencies 3–6 Hz. The observed increase in the mean clustering coefficient corresponds to more efficient information processing in the GSW network.

B3.Small-world propensity at 3 and 4 Hz

At 3 Hz, the Δ*_C_* and Δ*_L_* variables did not follow a normal distribution, unlike the SWP variable. The Wilcoxon signed-rank test was applied for the Δ*_C_* and Δ*_L_* and paired *t*-test for SWP. The medians of Δ*_C_* and Δ*_L_* differed statistically significantly between the two states, and more specifically, the median of Δ*_C_* decreased in the GSW state z = −3.49, *p* < 0.001. In contrast, the median of Δ*_L_* increased statistically significantly in the GSW state Z = −3.28, *p* = 0.001. For SWP, we observed a statistically significant increase in the mean value in the GSW state M = 0.067, SD = 0.12, T (22) = 2.6, *p* = 0.016. At 4 Hz, Δ*_C_* and Δ*_L_* did not follow a normal distribution, unlike SWP. The Wilcoxon signed-rank test was applied for the Δ*_C_* and Δ*_L_* and paired *t*-test for SWP. The medians of Δ*_C_* and Δ*_L_* differed statistically significantly between the two states, and more specifically, the median of Δ*_C_* decreased in the GSW state z = −3.74 *p* < 0.001. In contrast, the median of Δ*_L_* increased statistically significantly in the GSW state Z = −3.92 *p* < 0.001. Finally, regarding SWP, there was no statistically significant difference between the mean values of the RS state and the GSW state M = −0.037, SD = 0.14, T (22) = 1.25, *p* = 0.22. These differences are shown in Figure 4.

We demonstrated that at both frequencies, Δ*_C_* significantly decreased in GSW state, which indicated a similarity to the lattice network. Furthermore, at both frequencies, Δ*_L_* significantly increased in RS state, corresponding to divergence from the random network. Interestingly, the mean small-propensity values were marginally higher than 0.6, with a statistically significant increase in the GSW 3 Hz state. Furthermore, the importance of δ increased at the GSW state at both frequencies. This variation indicates the similarity of the clustering coefficient to its null model (lattice network). These results can be interpreted as a tendency of network regularization at the GSW state.

B4.Integrated value of influence at 3 and 4 Hz

It was observed that at 3 Hz, regardless of the state, the left anterior cingulate, left temporal pole, right fusiform gyrus, right gyrus rectus and left insula were influential nodes. At the GSW state, additional influential nodes were the right angular gyrus, right inferior parietal, left precentral gyrus, left parahippocampal gyrus and right middle temporal gyrus. Furthermore, at the GSW state at 4 Hz, the bilateral inferior and superior parietal gyri, precuneus, right superior temporal gyrus and the triangular part of the left inferior frontal gyrus had the higher IVI values. All nodes with an IVI value greater than the mean + 1 SD are shown in Table 3.

C.Connectivity patterns among influential nodes

The NBS analysis with a much stricter value of threshold (T = 3.5) and the utilization of the BrainNet Viewer Toolbox led to the visualization of connections among influential nodes during the GSW activity. Figure 2 shows connections presented with statistically significant probability comparing the GSW and RS states (threshold = 3.5, *p* < 0.001) at 3 Hz. At 3 Hz, the most statistically significant connections were observed between the right middle temporal gyrus and left superior temporal gyrus (T = 4.92, *p* < 0.001), left superior temporal gyrus and left precentral gyrus (T = 4.55, *p* < 0.001), and left insula and right fusiform (T = 4.81, *p* < 0.001).

Figure 3 shows the connections presented with statistically significant probability comparing the GSW and RS states (threshold = 3.5, *p* < 0.001) at 4 Hz. At 4 Hz, the most statistically significant connections were observed between the left inferior frontal gyrus, triangular part and right precuneus (T = 4.21, *p* < 0.001), and left inferior parietal gyrus (T = 3.58, *p* < 0.001); right superior temporal gyrus and right superior parietal gyrus (T = 3.76, *p* < 0.001), and left superior parietal gyrus (T = 3.87, *p* < 0.001).

## 4. Discussion

We investigated the EEG source localization of interictal GSWDs and functional connectivity in a cohort of 23 patients with GTCSa subsyndrome. Source localization analysis demonstrated that most spikes originated from the superior frontal gyrus and anterior cingulate, which have been reported as parts of the default mode network (DMN) [61,62]. These results add to the importance of the DMN in the epileptogenesis of GSWDs, which has also been suggested in a MEG study of patients with JAE [63]. Source analysis in patients with IGEs has been previously studied, utilizing different methods for spike localization. In absence epilepsy, the midpoint of the ascending phase of the spike was most frequently localized in the superior and middle frontal gyrus, while the spikes’ peak sources were in the sublobar, limbic and frontal lobes [64,65]. In patients with JME, a study revealed that the onset and peak of spikes were localized in sublobar regions and the frontal lobe, respectively [66].

Graph theory and cortical connectivity analysis were elaborated on networks with nodes which resulted from source localization. The network topology included anatomical areas which were not only bilaterally but also unilaterally represented [67]. In patients with GTCSa, there was an efficient flow of information across different nodes at the frequencies 1–12 Hz. Concerning the network’s segregation, it was influenced by GSW state segregation, mainly at 3–6 Hz. The SWP analysis revealed that the network deviated from a lattice and random network at the RS and GSW state, respectively. The network divergence during the GSW state indicated a tendency of network regularization. The resting-state connectivity analysis in EEG networks has been thoroughly reviewed and revealed that the small-worldness increased in comparison with normal subjects [67]. In contrast, concerning the EEG network’s behavior during GSWDs, there are a few studies. In patients with JME subsyndrome, GSWDs were correlated with increased long-range connections and reduced small-worldness [68]. On the other hand, there was an increase in the clustering coefficient and characteristic path length in patients with AE, which can be interpreted as the regularization of the network during ictal activity [69].

The IVI calculation showed that nodes such as the left fusiform and left anterior cingulate preserved their influential role during both RS and GSW state. The most influential nodes were not correlated with the most frequent sources of GSWDs. Furthermore, the left insula and left inferior parietal gyrus were the most influential nodes during GSWDS at 3 and 4 Hz, respectively. All influential nodes have been reported as parts of the DMN, with the exemption of the precentral gyrus, which belongs to the somatosensory network (SM). Thus, we concluded that the default mode network had a crucial role in the ictogenesis of GSWDs as well as in the influence on the source network. Lee et al. demonstrated that in patients with GTCSa subsyndrome, the anterior cingulate cortex and hippocampi, which belong to DMN, were crucial hubs during resting-state activity [26].

The connectivity pattern was investigated among the most influential nodes comparing RS and GSWD activity at 3–4 Hz. An intra-network connectivity of the aforementioned areas of DMN (Table 3) as well as internetwork connectivity between superior temporal and precentral gyri were statistically significant. The resting-state connectivity analysis in patients with IGEs revealed increased inter-hemispheric coherence in the θ band [70,71]. Furthermore, in patients with JME, cortical connectivity between cingulate gyrus-cuneus and paracentral lobule-cingulate gyrus in the pre-ictal epochs was increased compared to resting-state activity [72].

To our knowledge, this is the first low density-EEG cortical connectivity study in patients with GTCSa focused on SWP and IVI during GSW activity. The limitations of this study were the small number of patients, the absence of a control group and restrictions to the head modeling process’ validity. Furthermore, the EEG data were not preprocessed with an automated method for muscle artifact removal [31]. Epochs were selected through visual inspection of the EEG. The use of a template anatomy and the default MNI electrode positions restrained the accuracy of the head modeling process [73]. The low-density EEG decreased the sensitivity of the source localization method, especially with the sources which belonged to the limbic lobe [74]. Moreover, our patients were not drug naive. It has been demonstrated that antiepileptic drugs, such as valproic acid and lamotrigine, interfere with EEG connectivity [70,75]. Remarkably, the IVI method identified the left thalamus as the most influential node during RS activity at 3 Hz, consistent with studies that noted the significance of the thalami during RS activity [76,77]. On the other hand, the same method did not attribute an influential role to the thalami during GSW activity. We hypothesized that the calculation of mean cortical connectivity during the epochs did not reveal the crucial role in the epileptogenesis of GSW. Furthermore, it has been demonstrated in an EEG-functional MRI study that the thalami did not sustain the same level of activity during the generalized spike-wave complexes [23].

## 5. Conclusions

In our EEG network study, the source localization analysis with the sLORETA method concluded that the most common sources in epilepsy with GTCSa lay in the superior frontal gyrus and anterior cingulate. The imaginary part of coherence and graph theory metrics estimation revealed that the sources formed a network, which integrated information more efficiently during GSW at frequencies 1–12 Hz compared to RS. This network also had higher segregation during GSW at frequencies from 1 to 6 Hz and 10 Hz. Additionally, graph theory analysis indicated a tendency of regularization during GSW at 3–4 Hz. Finally, connectivity pattern analysis demonstrated an increased intra-network connectivity of DMN areas and inter-network connectivity of DMN and SM. Notably, some nodes acted mainly as generators of GSWDs and others as influential ones across the whole network. Future studies with a more accurate head modeling process (high density EEG, digitized electrode sensors coordinates and individual brain anatomy) are required to verify the participation of the limbic lobe in the epileptogenesis of GSWDs in GTCSa.

## Data Availability

The data that support the findings of this study are available from the corresponding author upon reasonable request.

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
