# Peer review of "EEG Network Analysis in Epilepsy with Generalized Tonic–Clonic Seizures Alone"

_brainsci, 2022, doi:10.3390/brainsci12111574_

Round 1

Reviewer 1 Report

The authors investigated the network that takes part in the formation of the spike-wave discharges in patients with Generalized Tonic-Clonic Seizures Alone and elucidate the network characteristics. They concluded that some nodes acted mainly as generators of GSWDs and others as influential ones across the whole network. This is a very interesting topic, a well written manuscript.

However, there were several major limitations for this study. First of all, there were no healthy subjects as a control group. Therefore, the functional network changes in the resting state between patients with GTC alone and healthy controls could not be compared. Second, although the electrodes were arranged according to the 10-20 system, the location of the electrodes placement of the actually attached sensor-level cannot be accurately known. Therefore, it was difficult to trust the results in the analysis using source-level electrode nodes based on inaccurate sensor level electrodes.

Reviewer 2 Report

The authors performed an EEG network analysis in patients with Generalized Tonic Clonic Seizures Alone (GTCSa). In order to localize the sources of generalized spike-wave discharges (GSWDs) the Standardized Low-Resolution Electromagnetic Tomography (sLORETA) method was used. Source localization revealed that most spikes arise from the superior frontal gyrus and anterior cingulate. Brain connectivity was computed by means of the imaginary part of coherence and the network properties were explored by graph theory analysis. The results suggested a tendency of network regularization during ictal activity and increased intra-network connectivity of default mode network areas and inter-network connectivity of default mode network and somatosensory network.

The study is interesting and can be considered a preliminary study useful for future investigations. 

I have the following comments and suggestions:

1) Why did you apply two band-stop filters? (Section 2.3.1) 

2) Add more details about the definition of the forward problem (section 2.4.1), inverse problem (section 2.4.2), brain functional connectivity (section 2.5) and graph theory (section 2.7).

3) Define all abbreviations before using them.

4) Add parameter definitions in all equations.

Reviewer 3 Report

Reviewer’s Report on the manuscript entitled:

EEG Network Analysis in Epilepsy with Generalized Tonic Clonic-Seizures Alone.

The authors defined a network for the formation of the spike-wave discharges in patients with generalized tonic clonic seizures alone (GTCSa) and elucidate the network characteristics. They found that GTCSa are the most common sources lay in the superior frontal gyrus and anterior cingulate. In my view the topic and results are sound, but the presentation can be improved. Literature review and conclusions can be improved. There are several places that require explanation. Please see below my comments.

Line 28. Please define IVI. Please define all the abbreviations the first time they are used. Please use all lower case when defining the abbreviation and be consistent with their style.

The Introduction and literature review can be improved. Since this research focuses on EEG network analysis, I suggest authors elaborate in a few paragraphs on EEG signals and images, sources of noise/artifacts, such as eye blink, and muscle movement, and applications in various fields, such as EEG for emotions classification, seizure, and sleep disorder. The following articles may also be included:

https://doi.org/10.3390/signals2030024

https://doi.org/10.3390/s22082948

https://doi.org/10.3390/s22062346

Line 98. How is this process done? What do you mean by the author? Is it you? If so, then please say “we” otherwise please say “specialist” instead of “author”

Line 109. Ocular movement. Please elaborate more here. Please say “specialist” instead of “author”. What technique is used to remove these artifacts? Please see the last two papers mentioned above.

Line 119. Please define all the parameters in this equation as well as other equations.

Line 142. How is an artifact-free epoch selected?

Lines 186-195. Style issue. Please have the equations written in a separate line. Please also define all their parameters.

Line 407-408. Please merge it into the next paragraph. Please note that a paragraph needs to have at least two sentences.

The Conclusions can be improved. Please state the objective, method, result, area of improvement, and future work there.

Regards,

Round 2

Reviewer 1 Report

Good work. I think it's a good study.